# Virus-Like Particle-Mediated Vaccination against Interleukin-13 May Harbour General Anti-Allergic Potential beyond Atopic Dermatitis

**DOI:** 10.3390/v12040438

**Published:** 2020-04-13

**Authors:** John Foerster, Aleksandra Molęda

**Affiliations:** Department of Molecular and Clinical Medicine, University of Dundee, Medical School, Ninewells Hospital, Jacquie Woods Centre, Ninewells Drive, Dundee DD1 9SY, UK

**Keywords:** VLP, IL-13, interleukin-13, vaccine, Tfh cells

## Abstract

Virus-like particle (VLP)-based anti-infective prophylactic vaccination has been established in clinical use. Although validated in proof-of-concept clinical trials in humans, no VLP-based therapeutic vaccination against self-proteins to modulate chronic disease has yet been licensed. The present review summarises recent scientific advances, identifying interleukin-13 as an excellent candidate to validate the concept of anti-cytokine vaccination. Based on numerous clinical studies, long-term elimination of IL-13 is not expected to trigger target-related serious adverse effects and is likely to be safer than combined targeting of IL-4/IL-13. Furthermore, recently published results from large-scale trials confirm that elimination of IL-13 is highly effective in atopic dermatitis, an exceedingly common condition, as well as eosinophilic esophagitis. The distinctly different mode of action of a polyclonal vaccine response is discussed in detail, suggesting that anti-IL-13 vaccination has the potential of outperforming monoclonal antibody-based approaches. Finally, recent data have identified a subset of follicular T helper cells dependent on IL-13 which selectively trigger massive IgE accumulation in response to anaphylactoid allergens. Thus, prophylactic IL-13 vaccination may have broad application in a number of allergic conditions.

## 1. Introduction: Why Develop an Anti-IL-13 Vaccine?

In an age dominated by an avalanche of monoclonal antibodies (MAb) hitting the market for a dizzying variety of conditions, one may ask why replacing this obvious successful model with a vaccine strategy is sensible at all. The answer to this question is four-fold. Two of these are generic and would apply to many vaccines replacing a MAb treatment. First, it will be much more accessible to patients (economics). Second, it will have much broader applications thanks to the qualitatively different nature of a polyclonal vs. a monoclonal immune response (see below for details). The third reason is more limited to IL-13: it has an excellent safety profile as a target compared to almost all other cytokines, with the exception of IL-17. Fourth, and finally, very recent data suggest that targeting IL-13 with a VLP vaccine may have very broad anti-allergic potential, possibly leading to amelioration of allergies and being able to achieve synergistic effects with other anti-allergic treatments. The present review will discuss these four themes in turn, not aiming to be exhaustive, but rather with the intent to stimulate further efforts to tackle many open questions. By implication, therefore, this review does not intend to deliver a broader review of the signalling, structure, or immunobiology of IL-13 which have been reviewed excellently recently elsewhere [1,2,3].

## 2. Virus-Like Particles as a Construct for IL-13 Therapeutic Vaccine

Virus-like particle (VLP) vaccines are nanostructures generated by self-assembly of structural proteins resembling the native version in their morphology and composition but lack the genomic material of infectious capacity [4,5]. VLPs can be acquired from a variety of expression systems and platforms including bacteriophages (MS2, PP7 and AP205 [6,7,8]), yeast (*Hansenula polymorpha* and *Saccharomyces cerevisiae* [9,10]), bacteria (*Escherichia coli* [8]), mammalian cell lines (Vero, 293T and BHK cell lines [11,12,13]), plant cell culture (cowpea mosaic virus, cucumber mosaic virus, tobacco mosaic virus, and bean yellow dwarf virus [14,15,16]) and insect cell lines (Baculovirus and Sf9 cell line [12,17]) [18]. Vaccine development faces a clear challenge: production of sufficient amounts of quality antibodies to target the desired antigen. VLPs provide an excellent vaccine delivery platform due to their composition: their small size (usually 20–200 nm), geometry and flexibility during development [4]. Their size allows easy passage and drainage through the lymph to reach all areas such as secondary lymphoid organs resulting in profound effects in targeting follicular B cells [4,19,20,21]. Furthermore, CD8+ and plasmacytoid subsets of dendritic cells (DCs) can cross-present small-sized antigens such as VLPs and active B cells and T cells in the lymph nodes to induce cytotoxic effects [20,22,23]. Repetitive multivalent surface arrangement allows cross-linking of B cell receptors, perfect for inducing great amounts and long-lasting antibodies [20,24]. VLPs also act as a template for further engineering, where additional epitopes, proteins and nucleic acids are easily incorporated alongside vaccine targets that can significantly increase immunity such as Toll-like receptor (Tlr) ligands [20,25]. These characteristics can thus provide solutions for vaccine delivery challenges and are readily modified for a vast variety of constructs to boost immune responses in many individuals.

## 3. The Health Economics of IL-13-Targetable Diseases

In terms of economics, it is obvious that health care systems globally are under huge strain; personal bankruptcies due to health care expenditure in the US alone tell the story: a 2019 study in the American Journal of Public Health found that two-thirds of personal bankruptcies are filed due to medical bills, equating to more than half a million of affected people despite the Affordable Care Act [26]. While health care cost in other economies may not be quite as exorbitant, that fact is offset by the simple unavailability of many high-quality medicines to patients who cannot afford private health care. Given current demographic trends toward increased old age-related morbidity, including the dementia ‘epidemic’, as well as globally increased longevity, the search for truly affordable health care solutions represents a distinct priority.

The clinical indications amenable to anti-IL-13 vaccination based on documented action of anti-IL-13 MAbs to date include atopic dermatitis, subgroups of asthma, and eosinophilic esophagitis. However, the list of other potential indications is much longer and has been discussed in detail [27]. Crucially, in the context of competitive resource allocation vis-à-vis conditions such as dementia, cancer, and emerging infectious diseases, it is clear that per-case expenditure available by health care providers will not be able to satisfy the profit margins required to offset large-scale manufacture of monoclonal antibodies. Hence, vaccine approaches, which also avoid the need for laboratory monitoring infrastructure, will become eminently competitive in the near future.

## 4. Monoclonal Antibodies vs. Polyclonal Vaccine Responses

The rate of recent marketing approvals of MAbs and sometimes decoy receptors suggests that they are highly effective in ameliorating disease. However, a closer look prompts the question: do they reach full therapeutic potential? Specifically, it is becoming increasingly evident that the serum concentrations required for monoclonal antibodies to be effective are rather extreme. A striking example for this is the group of monoclonals targeting the p19 subunit of IL-23, currently licensed for psoriasis: guselkumab, risankizumab, and tildrakizumab. While the molecular mode of action is identical between all three antibodies, even the comparatively high affinity of tildrakizumab to its cytokine target (300 pM) is evidently suboptimal based on its inferior clinical activity compared to the competitor MAb, which feature Kd values in the vicinity of a staggering 2 pM (Table 1). Notably, the relatively low efficacy of tildrakizumab exists despite a much higher relative affinity of this MAb for the cytokine compared to the receptor (Table 1).

For MAbs targeting the IL-4/13 system, this overall scenario appears similar. MAbs and their respective molecular targets are graphically summarised in Figure 1. Structural and signalling details have been summarised in detail recently elsewhere [2]. As detailed in Table 2, the picture emerging from an integrated review of clinical success (or failure) of related MAbs is that affinities of Kd values of approximately 60 pM or higher appear to correlate with clinical activity, justifying clinical development all the way through phase III trials. The failure of pascolinumab is mechanistically of interest, as this MAb harbours high affinity. By implication, its lack of satisfactory activity in clinical trials would clearly suggest that isolated IL-4 inhibition, with intact IL-13 signalling, is insufficient to achieve useful suppression of pro-inflammatory signalling.

The apparent requirement for ultra-high cytokine affinity of MAbs is all the more noteworthy, as trough serum concentrations are usually above 5 g/mL. By contrast, serum concentrations reported for IL-13 range from below 1 pg/mL, regardless of whether in healthy probands or symptomatic asthma patients [48], to between 0.16 and 0.24 ng/mL in diverse populations and patient groups [49]. Even if concentrations in target tissues (lung or skin) may be significantly higher than in serum, it is remarkable that molar excess of several orders of magnitude of MAb to cytokine still can prove clinically insufficient.

By contrast, polyclonal responses act by combining various affinities of different antibody species. The quantity of B cell clones, as well as resulting affinities, can vary widely. Thus, a bivalent VLP-based vaccine against norovirus consisting of Gll.1 and Gll.4 sequence strains was found to produce an oligoclonal response, with < 3 antibodies accounting for 58–86% of epitope-specific clones [14]. This study utilised the VEE replicon vector to transfect ORF2 genes of norovirus for production of viral protein 1, a capsid protein that can self-assembly into a VLP when introduced into BHK-21 cells. By contrast, a recent in-depth analysis showed that influenza virus vaccination in humans triggered 36 ± 12 (mean ± SD, range 16–49) B cell lineages expanding >50-fold and, collectively, accounting for 22 ± 12% of each subject’s antibody repertoire during peak response [50]. In another influenza vaccine study, vaccination elicited between 40 and 147 clonotypes where the top 6% most abundant clonotypes accounted for > 60% of the entire repertoire [51]. In terms of affinity, Kd values between 2.5 pM and 160 nM were found [51]. A large-scale analysis of influenza vaccination in healthy controls and SLE patients elicited antibody affinities between 1 and 100 nM [52]. A single-protein vaccination of HIV-Env produced final affinity-matured Ig affinities to the antigen between approximately 3 and 10 nM in Rhesus monkeys [53]. Collectively, these data strongly suggest that vaccine-triggered natural effector antibodies show significantly less single-clone affinity compared to the engineered MAbs detailed in Table 1 but bear the potential of significant cooperative action. Although no detailed studies are available on therapeutic VLP vaccines, these likely trigger responses comparable to anti-virus vaccines. Accordingly, the VLP-based vaccines targeting interleukin-1 and angiotensin both met their pharmacodynamic endpoints in clinical trials [54,55]. Likewise, a number of VLP-based vaccines have been shown to be effective in other mammalian species, including treating insect bite hypersensitivity in horses [56]. Here, IL-5 was chemically conjugated to a VLP from cucumber mosaic virus engineered with the tetanus toxoid (TT) epitope, a property to enhance immunological responses of T helper cells and memory B cells [56]. This characteristic was also employed in vaccine constructs against psoriasis, cat allergy and Alzheimer’s disease [57]. Designing a VLP vaccine that contains the TT epitope alongside the target antigen can be a useful tool in ensuring enhanced immune protection across majority of immunised individuals, especially those that are aging [57]. Similarly, for influenza vaccines, polyclonal approaches may outperform monoclonal targeting based on broader antigen coverage [58].

Apart from the affinity and polyclonality of a vaccine response, a third fundamental difference between MAbs and polyclonal responses is that MAbs are of a singular—mostly IgG1—isotype, which is efficient in engaging effector functions such as antibody-dependent cytotoxity (ADCC), antibody-dependent cellular phagocytosis (ADCP), or pro-inflammatory cytokine activation, mediated by Fc binding of a variety of Fc gamma receptors (FcγRs). It is noteworthy that dupilumab is of the IgG4 isotype, which is less efficient in engaging FcγRs [59]. By contrast, polyclonal responses produce a mix of isotypes, which will vary between vaccine recipients in addition to cooperative action between various species targeting the cytokine at different epitopes. 

## 5. The Inhibition of Anaphylactic TfH Cells: A Novel Role for IL-13 Neutralisation

Very recently, the paradigm in IL-4/IL-13 immune function in the germinal centre reaction was that IL-4 is central to germinal centre-based affinity maturation, as well as isotype switching toward IgE of B-cells, but not IL-13, through follicular T-helper (TfH) 2 cells characterised by IL-4 secretion (reviewed in [60]). However, a very recent landmark study identified a new Tfh subset characterised by IL-13 secretion which is responsible for the generation of IgE^high^ B cells in response to stimulation with allergens [61] (Figure 2). Crucially, TfH13 cells were not only identified in the newly created hyper-IgE-harbouring mouse model, but also detectable and increased in humans with peanut allergy. Importantly, TfH13 cells did not drive IgE^high^ B cell and peripheral IgE production in response to parasitic (helminth) infection but only in response to anaphylactoid allergen challenge (also reviewed in [62]). Moreover, T cell-specific ablation of IL-13 was found to profoundly suppress IgE synthesis. These data suggest that elimination of IL-13 function may lead to suppression of anaphylactic reaction to a wide range of allergens.

As detailed above, the response to such therapeutic intervention may profoundly differ between anti-IL-13 MAb administration and anti-IL-13 vaccination, in particular since the latter is expected to induce IL-13-targeted memory clones (both B and T cells), which could modulate the germinal centre reaction by cytokine secretion and which would not feature in MAb-mediated IL-13 suppression. In addition, it is intriguing that TfH1 cells, which would mediate class switching in response to a VLP-type vaccine, would be instrumentalised in this way to suppress the activity of TfH13 cells. In fact, this concept appears to already have been validated for TfH2 cells, since blocking of IL-4 caused a profound block in Th2-class switch and improved antibody responses to oxycodone and diptheria-pertussis-tetanus vaccination [64]. Clinically, it is very possible that the routinely observed major drop in serum IgE levels in patients treated with dupilumab is in part mediated by suppression of TfH2 cells. It would therefore be of interest to compare the performance of monoclonal- vs. vaccine-mediated IL-13 inhibition in the novel DOCK8-/- mouse model established in the above cited study. Moreover, given the different mode of action of a polyclonal response, combined with the novel anti-TfH13 mode of action, it is entirely possible that VLP vaccine-based IL-13 targeting will perform different to MAb-based treatment in allergic asthma.

## 6. Safety of Anti-IL-13 Vaccination

In light of the above-detailed roles of IL-13 and IL-4, it is both highly significant and reassuring that a large double-blind controlled study on almost 180 patients did not find any effect of tetanus vaccine-induced specific IgG formation after twelve weeks of dupilumab treatment [65]. Furthermore, emerging post-marketing surveillance data of patients on dupilumab confirms an approximately 10% incidence of conjunctivitis. In one recent real-world review of 54 patients, 2 patients had to discontinue due to, or for reasons related to, conjunctivitis symptoms [66]. Since this drug-specific side effect had already been recognised during pre-marketing phase II–III studies and continues to be non-reported in any anti-IL-13 treated cohort, it is highly likely that conjunctivitis will not be associated with long-term IL-13 elimination. Furthermore, a growing array of meta-analyses, and systematic reviews of anti-IL-13 MAb-treated patients fail to uncover target-related limiting adverse events in large patient groups [27,67,68,69].

## 7. Conclusions

Utilising VLPs as vaccine constructs provides flexibility and many prospects in design that can significantly enhance protection and immunogenicity. IL-13 represents an excellent proof-of-concept target to validate anti-cytokine vaccination as a therapeutic strategy in humans based on validated vaccine targets, an excellent target safety profile, and broad vaccine applicability. Treating IL-13-related immunological disorders with a therapeutic vaccine will offer a new avenue in medical intervention.

## Figures and Tables

**Figure 1 viruses-12-00438-f001:**
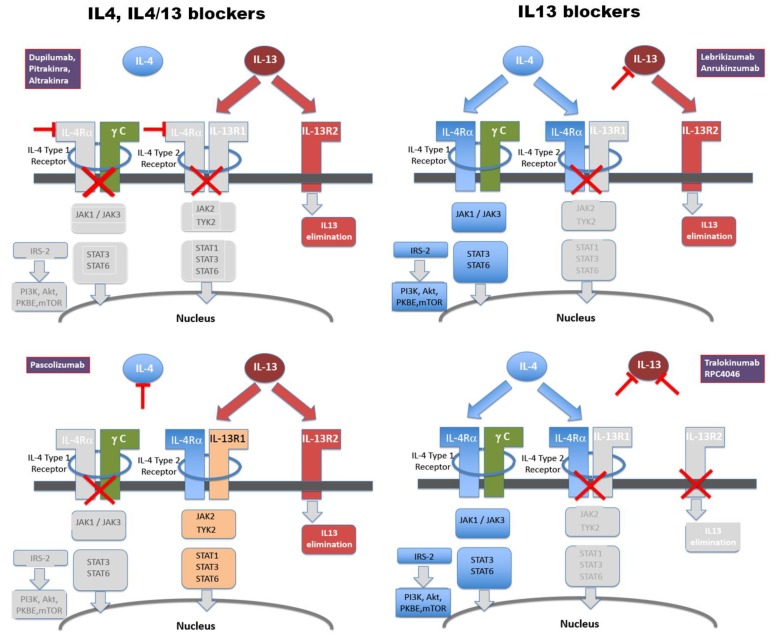
Molecular targets of monoclonal antibodies clinically evaluated to inhibit IL4/IL-13 signalling to date. For a list of antibodies and references, see Table 1. Colour codes used are blue for signalling transduced through IL-4, but not IL-13, beige for either IL-4 or IL-13 (through the IL-4Ra/IL-13Ra1 receptor dimer), and red for IL-13, but not IL-4. IL-13Ra2, also depicted in red, acts primarily as a cellular disposal facility for IL-13 [34]. For details on structure and signalling, see 2.

**Figure 2 viruses-12-00438-f002:**
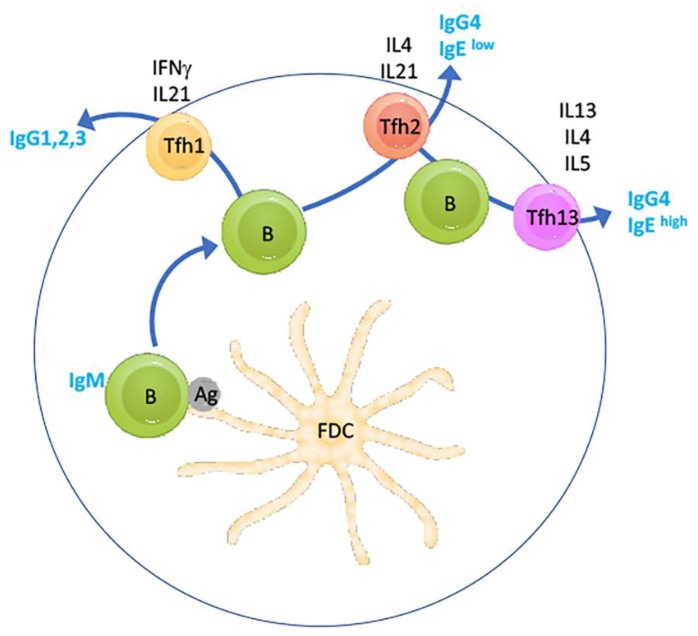
Function of IL-13-dependent TfH cells in IgE production in response to allergens. Simplified schematic showing the light zone of a peripheral lymphoid germinal centre (GC) where a naïve IgM-positive B cell survives apoptosis after antigen (Ag) encounter presented by a follicular dendritic cell (FDC). Prior and additional interaction with Th cells are not shown. Depending on the type of antigen, the B cell may then receive stimulation by TfH1 cells, resulting in class switching to Ig-isoforms, as indicated. Further, sequential stimulation by TfH2 cells may cause secondary class switching to IgG4 or IgE^low^ phenotypes. Finally, the newly identified IL-13-dependent TfH13 cells, may cause tertiary switching to IgE^high^ in the context of allergen stimulation. The schematic is based on figures in [61,63].

**Table 1 viruses-12-00438-t001:** Association of IL-23-targeting monoclonal antibody (MAb) affinity with clinical efficacy.

	Target MAb Affinity	Clinical Efficacy ^1^	Fold-Increase Vs. Receptor Binding ^2^	Ref
Guselkumab	2 pM	91%	3000-fold	[28,29]
Risankizumab	2 pM	91%	3000-fold	[30,31]
Tildrakizumab	300 pM	61%	20-fold	[32]

^1^ Response of psoriasis severity, measured as the so-called PASI75 index, indicating 75% improvement from baseline, measured at twelve weeks after treatment start, in each study cited. ^2^ Compared to the 6 nM reported affinity of IL-23 to receptor [33].

**Table 2 viruses-12-00438-t002:** Ligand affinities and affinities relative to ligand–receptor binding of various cytokine-binding clinically used MAbs.

Mab/Blocker	Net Target ^1^	Affinity (pM)	Tested in ^2^	Effective in	Fold Affinity Vs. Receptor ^3^	Ref
Group 1: IL-13-only blockers						
Lebrikizumab	IL-13Ra1	<10	Asthma, AD	AD, Asthma ^4^	3.5	[35,36]
Anrukinzumab	IL-13Ra1	385	UC, Asthma	---	0.1	[37]
RPC4046	IL-13Ra1/Ra2	50	EoE	EoE	0.7	[38,39]
Tralokinumab	IL-13Ra1/Ra2	58	Asthma, AD	AD	0.8	[40]
Group 2: IL-4-selective and IL4/IL-13 combined blockers						
Pascolizumab	IL-4Ra	60	Asthma	---	1.4	[41]
Dupilumab	IL-4Ra, IL-13Ra1	9	Asthma, AD	Asthma, AD	10	[42]
Pitrankinra	IL-4Ra, IL-13Ra1	100 ^5^	Asthma	---	1	[43]
Altrakincept	IL-4Ra, IL-13Ra1	1000	Asthma	---	0.1	[44,45]
AMG317	IL-4Ra, IL-13Ra1	180	Asthma	---	0.8	[46]

^1^ Signal-transducing receptor pathway inactivated by monoclonal antibody. The actual targets of each IL-13- and IL-4-inhibiting MAb are shown in Figure 1. ^2^ Disease abbreviations: UC—ulcerative colitis, EoE—eosinophilic esophagitis, and Pso—psoriasis. ^3^ Target–receptor affinities are complex, involving sequential ternary complex formation of more than one receptor subunit. The data shown depict the ratio between MAb/cytokine affinity and cytokine/receptor affinity, respectively, for IL-13 -> IL-13Ra1/IL-4Ra (30–40 pM, for group 1), and for IL-4 -> IL-4Ra (100 pM, group two), as detailed in [47]. The highest affinity target/receptor binding reported was chosen as most relevant to gauge the competitive strength of drug circulating in serum. ^4^ In the high-biomarker patient subgroup. ^5^ Assuming retained affinity for the dual mutation differentiating pitankinra from native IL-4.

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
