# Peer review of "Virus-Like Particle-Mediated Vaccination against Interleukin-13 May Harbour General Anti-Allergic Potential beyond Atopic Dermatitis"

_viruses, 2020, doi:10.3390/v12040438_

Round 1

Reviewer 1 Report

I was not able to judge what is a citation from what is a conclusion from the author since the references used are mostly absent in the text or not properly formatted

The title is misleading, two thirds of the paper is about monoclonals and their molecular targets.

What type of VLPs can be used? what expression systems should be used? Why vlps and not other types of vaccines (m-RNA, peptides…). The topic is quite interesting, but it seems poorly supported.

The conclusion section should be extended, it simply lacks the punch.

Author Response

Query 1: I was not able to judge what is a citation from what is a conclusion from the author since the references used are mostly absent in the text or not properly formatted

The title is misleading, two thirds of the paper is about monoclonals and their molecular targets.

What type of VLPs can be used? what expression systems should be used? Why vlps and not other types of vaccines (m-RNA, peptides…). The topic is quite interesting, but it seems poorly supported.

The conclusion section should be extended, it simply lacks the punch.

Reply:  Thank you very much for your feedback on our manuscript. We would like to express our gratitude for your time taken for a thorough reading of the manuscript. To address raised queries:

The referencing and formatting of the paper have been corrected to ensure the manuscript is clear to follow with the conclusion amended. We have enriched the review by adding a paragraph (Virus-like particles as a construct for IL-13 therapeutic vaccine, lines 40-60) on types of VLPs, expression systems and why VLPs are an attractive vaccine platform to provide better understanding for the reader.

Reviewer 2 Report

This article reviews the use of active immunizations targeting IL-13 for anti-allergic diseases. The title "virus-like-particle mediated vaccination..." but the main text has not specifically explain what and why the use of VLP-mediated  immunizations to target the IL-13 diseases. Also the authors stated the norovirus VLP vaccination and the horse VLP vaccine but have not detailed the VLP constructs and their distinctive elicitation of the targeted immune responses . Similar article(s) has been published for a plant virus-based VLP against IL17. This review should include the details of VLP constructs if the authors use "VLP..." as the title.

Author Response

Query 2: This article reviews the use of active immunizations targeting IL-13 for anti-allergic diseases. The title "virus-like-particle mediated vaccination..." but the main text has not specifically explain what and why the use of VLP-mediated  immunizations to target the IL-13 diseases. Also the authors stated the norovirus VLP vaccination and the horse VLP vaccine but have not detailed the VLP constructs and their distinctive elicitation of the targeted immune responses . Similar article(s) has been published for a plant virus-based VLP against IL17. This review should include the details of VLP constructs if the authors use "VLP..." as the title.

Thank you very much for your feedback on our manuscript. We were gratified to see that the reviewer appeared overall to be satisfied with the general quality of the paper.

We have addressed the queries by including a new paragraph (Virus-like particles as a construct for IL-13 therapeutic vaccine, lines 40-60) to explain why VLPs should be used for vaccine delivery for an anti-IL-13 immunisation. We are also pleased to include information on the norovirus vaccine (lines 129-134), treating insect bite hypersensitivity in horses with IL-5 VLP vaccine (lines 147-151), and other VLP constructs (lines 151-156).